# Beyond CheatBots: Examining Tensions in Teachers' and Students' Perceptions of Cheating and Learning with ChatGPT

Christopher Mah [1,*], Hillary Walker [2], Lena Phalen [1], Sarah Levine [1], Sarah W. Beck [3] and Jaylen Pittman [1]

1 Graduate School of Education, Stanford University, Stanford, CA 94305, USA; lphalen@stanford.edu (L.P.); srlevine@stanford.edu (S.L.); jaylen@stanford.edu (J.P.)
2 Berkeley School of Education, University of California, Berkeley, CA 94720, USA; hillaryewalker@berkeley.edu
3 Department of Teaching and Learning, New York University, New York, NY 10003, USA; sarah.beck@nyu.edu
* Correspondence: chrismah@stanford.edu

**Abstract:** As artificial intelligence (AI) is increasingly integrated into educational technologies, teachers and students must acquire new forms of AI literacy, including an understanding of responsible use of AI. In this study, we explored tensions in teachers' and students' opinions about what constitutes learning and cheating with AI. Using qualitative methods, we asked Pre-K through postsecondary writing teachers (n = 16) and a linguistically diverse group of students (n = 12) to consider examples of how students might use ChatGPT, rank them in order of how much they thought each student learned and cheated, and explain their rankings. Our study yielded three findings. First, teachers and students used similar criteria to determine their rankings. Second, teachers and students arrived at similar conclusions about learning with ChatGPT but different conclusions about cheating. Finally, disagreements centered on four main tensions between (1) using ChatGPT as a shortcut versus as a scaffold; (2) using ChatGPT to generate ideas versus language; (3) getting support from ChatGPT versus analogous support from other sources; and (4) learning from ChatGPT versus learning without. These findings underscore the importance of student voice in co-constructing norms around responsible AI use.

**Keywords:** ChatGPT; writing; cheating; plagiarism; AI literacy; generative AI; teacher perceptions; student perceptions; qualitative research; comparative study

## 1. Introduction

### 1.1. Background

Educators have responded to rapid advances in generative artificial intelligence (AI) with both enthusiasm and fear. Some see the potential for new text generation tools like OpenAI's ChatGPT to serve as a virtual teaching assistant or personalized tutor [1]. Others are alarmed by concerns about misinformation, bias, and privacy [2]. In their systematic review of ChatGPT use in K-12 education, Zhang and Tur [3] identified potential threats to students, including possible violations of students' data privacy, increased opportunities for academic dishonesty, and diminished student engagement. Teachers and administrators have expressed particular concerns about students using these tools to cheat and plagiarize [4]. Even though some students do misuse AI in ways that constitute cheating, some administrators and teachers may make deficit-based assumptions about students' intentions and motivations. Additionally, a history of gender and racial disparities in student discipline [5,6] suggests that teachers may act on discriminatory perceptions about which students are most likely to use AI to cheat. To avoid unfair policies and enforcement, and to ensure every student develops the skills they need to use AI productively, educators must first establish a shared discourse about AI, academic integrity, and learning.

Attempts to define AI literacy predate ChatGPT. Long and Magerko [7] define AI literacy as, "a set of competencies that enables individuals to critically evaluate AI technologies; communicate and collaborate effectively with AI; and use AI as a tool online,

at home, and in the workplace". Other researchers have avoided definition and instead proposed broad frameworks for AI literacy. Ng et al. [8] reviewed thirty peer-reviewed articles and identified four aspects for fostering AI literacy: understanding AI concepts, applying computational thinking, designing and creating with AI, and attending to AI ethics. Similarly, Heydar and Possega [9] characterized AI literacy as having three dimensions: functional/technical literacy, critical/cognitive literacy, and sociocultural/socioemotional literacy. Irrespective of conceptualization, defining responsible use of AI is a crucial part of any framework for equitable AI literacy [10]. Although technological advancements in education are often designed with equity in mind, they can instead reproduce or exacerbate educational inequity [11]. Technological leaps require new forms of literacies; however, schools often lack the resources, personnel, or capabilities to keep pace with change. Teachers and students rarely participate in developing policies and norms for adopting new learning technologies; yet, they are most directly impacted by them. Against this backdrop, there is a growing necessity for teachers and students to construct shared norms around responsible use of AI [12].

### 1.2. Historical Definitions of Responsible Technology Use in Schools

Schools have historically had difficulty defining "cheating", resulting in a wide range of policies around technology use [13,14]. Craig and Evans [15] found that teachers and students struggled to agree on a definition for plagiarism, with particular confusion noted between teachers and students as well as amongst teachers about whether paraphrasing or using work from another class constituted plagiarism. One reason that cheating is difficult to define is that different activities have different learning objectives. Dick et al. [16] proposed that a behavior can be defined as cheating if (a) it violates the rules that have been set for the assessment task, or (b) it violates the accepted standard of student behavior at the institution. Yet, they add, "In both cases, this assumes that the accepted rules and standards have been clearly laid out for students" (p. 172). Even when schools codify policies, students may not be aware of them. In a study of three thousand university students, Gullifer and Tyson [17] found that only around half of the students reported reading plagiarism policies.

Educational stakeholders have advanced competing ideas around responsible use of digital technology in classrooms and what constitutes cheating. As a result, many schools and districts have adopted the International Society for Technology in Education (ISTE) [18] standards, which provide general guidance for technology use, as well as more local, adaptive policies for plagiarism and technology. The ISTE released versions of their standards in 1998, 2007, and 2016 as technology rapidly evolved from overhead projectors to PowerPoints to smartphones and social media. The ISTE framed the responsible use of technology as a discussion of "digital citizenship" [1]. Specifically, the ISTE defines intellectual property as "content or ideas created by an individual or identity", and responsible use of intellectual property as "abiding by copyright and fair use, citing resources, gaining or giving permission to use (content), avoiding plagiarism, understanding and using creative commons" [19]. Other organizations, including the U.S. Department of Education, have adopted these definitions and ideas about digital citizenship [20].

### 1.3. Responsible Use in the Era of Generative AI

In the era of generative AI, schools have had even more difficulty defining responsible use of technology. Mandated distance learning during the pandemic and the release of ChatGPT in November 2022 have forced schools and individual educators to again reevaluate their policies around what constitutes original writing and responsible digital citizenship [21]. The use of ChatGPT further complicates what might be considered plagiarism, since AI-generated writing cannot be attributed to a specific primary source, forcing educators to reconsider, from an ethical standpoint, what it means to generate original content [22]. Without a uniform understanding and definition of key concepts such as "plagiarism", there is no way to manage it effectively [22]. Compounding the challenge

of defining plagiarism and cheating, ChatGPT writing is also largely undetectable by existing AI classifiers. In a comparison of five AI detection tools, including OpenAI's own, Chaka [23] concluded that none of the five detectors could "accurately and convincingly" detect AI-generated text.

Schools have adopted different approaches to meeting these new challenges. Some have implemented outright bans on ChatGPT, blocking it from their networks [22]. In contrast, scholars have urged schools to develop programs and curricula to teach students effective use of generative AI tools [24]. The ISTE has issued new programs for and examples of effective AI use in the classroom that embrace the technology's affordances and teach about its constraints [19], and the 2024 National Education Technology Plan (NETP) added the category of "algorithmic literacy" to the framework of "digital citizenship" for schools to adopt and implement [20]. Others have emphasized the importance of teaching students proper citation and attribution when writing with ChatGPT [25,26].

More broadly, scholars and educators have pointed to the rise of generative AI to call for reimagining assessment and writing assignments beyond multiple choice, definitions, problem sets, and the five-paragraph essay [27,28]. Some educators view this shift with ambivalence; in a widely circulated article in *The Atlantic* titled "The End of High School English", Herman [29] asked, "Is this moment more like the invention of the calculator, saving me from the tedium of long division, or more like the invention of the player piano, robbing us of what can be communicated only through human emotion?" Others view the shift as necessary and productive. For example, Rudolph et al. [27] conducted a literature review of 166 sources about ChatGPT (including peer-reviewed academic articles, conference papers, newspapers, and social media posts) and concluded:

> We advise against a policing approach (that focuses on discovering academic misconduct, such as detecting the use of ChatGPT and other AI tools). We favour an approach that builds trusting relationships with our students in a student-centric pedagogy and assessments for and as learning rather than solely assessments of learning [27,30,31].

While scholarship around responsible AI use in classrooms is growing, much of this work is focused on higher education and developed without input from teachers and students. This paper elevates the perspectives of teachers and students to inform the development of practical and equitable frameworks for responsible AI use.

## 2. Materials and Methods

### 2.1. Objectives and Research Questions

Our team investigated two separate groups: (1) teachers participating in a professional development institute on writing; and (2) linguistically diverse high school students. We wanted to explore teachers' and students' perceptions of learning and cheating with ChatGPT. Although many possible dimensions comprise responsible use, our team focused on the degree to which a student might use AI to learn or to cheat. We also limited our focus to one instantiation of generative AI, ChatGPT, because of its ubiquity in schools and its relevance to writing. We aimed to answer two questions:

> Question 1: *What criteria did these teachers and students use to define learning about writing with ChatGPT?*

> Question 2: *What criteria did these teachers and students use to define cheating at writing with ChatGPT?*

Following Saldaña [32], we used fundamental qualitative methods, relying on group discussion grounded in a textual artifact to elicit participants' ideas around the topic of learning and cheating with AI.

### 2.2. Participants and Context

### 2.2.1. Student Participants

The students in this study attended a small, private charter high school in a small city on the West Coast. According to the school's website, minority enrollment is 100%, with Latinx students making up the largest racial demographic (77%), and 97% would be the first in their family to attend college. We worked with an English teacher to recruit twelve students: one freshman, five sophomores, five juniors, and one senior. These students represented the broader school demographics and varied in their English class grades. Some were familiar with ChatGPT and others had never heard of it. Students voluntarily participated in the study, which consisted of four sixty-minute sessions after school. During these sessions, we asked students to write in different genres, alone and in pairs, with and without the assistance of ChatGPT. The data for this study is only from the first session, which we describe in the next section. We compensated students with gift cards.

### 2.2.2. Teacher Participants

The teachers in this study were participants in a three-week summer writing institute facilitated by a local chapter of the National Writing Project, an organization that provides writing teachers with professional development, networking, and leadership opportunities. Each summer, through a process of applications and community interviews, the organization selects a cohort representing a range of grade levels, teaching contexts, content areas, and teaching experience. The institute activities include lesson demonstrations by teacher fellows, book study and writing groups, racial affinity groups, and opportunities to engage in social and collegial conversations with other educators.

We worked with fifteen teacher fellows and one staff member who was also a teacher. We hosted sessions on Zoom and in person at the public university hosting the organization. The group included ten middle and high school teachers, three elementary school teachers, three post-secondary teachers, and one pre-K teacher. According to participants' self-reports on applications, fifteen of the sixteen participants were women, and ten were people of color. The professional and demographic diversity of this group allowed for diverse perspectives on cheating and learning. Thus, we used purposive sampling for maximum variation [33]. Additionally, because our study focused on ChatGPT as a writing tool, we worked with writing teachers who could talk about experiences relevant to our research questions in "an articulate, expressive, and reflective manner" [34]. We compensated all participants with gift cards for their time and intellectual labor.

### 2.3. Data Collection and Analysis

In this study, we sought to understand what criteria one group of teachers and one group of students used to define cheating and learning with ChatGPT. To achieve this, we designed an activity in which hypothetical students used ChatGPT in four different ways. We created examples to represent different uses of ChatGPT, including asking ChatGPT to suggest possible first sentences for an essay, create a content-specific outline, edit a completed paragraph, and suggest counterarguments. Then, we asked participants to work in pairs and rank the examples, first in order of how much each student learned and then in order of how much they cheated. We used a ranking task to prompt thoughtful reasoning and encourage participants not to "shortcut the response process" as they might do with a less cognitively demanding task, such as using a rating scale [35]. Finally, we asked them to discuss their rankings and provide rationales.

### 2.3.1. Student Data Collection

During the first after-school session, we presented students with a paper handout we had created. The handout showed screenshots of four different prompts that hypothetical students could give to ChatGPT to help them write an argumentative essay, as well as ChatGPT's responses. The prompts were:

Student A: *Give me two possible first sentences for an essay arguing that schools should not require students to wear uniforms.*

Student B: *Make an outline with only headers and subheaders for me to use as I write an essay arguing that schools should not require students to wear uniforms.*

Student C: *Fix this paragraph I wrote arguing that school uniforms discourage students' individuality.* (*This prompt included a pre-written paragraph with grammatical and spelling errors).

Student D: *I am going to write an argument where I argue against mandatory school uniforms, but I am having trouble writing it. Please help me write my argument by taking the other side and giving me a list of counterarguments.*

We asked students to review the examples and to rank them, first in order of how much they thought each student learned and then in order of how much they thought each student cheated. Then, we asked them to discuss their rationale as thoroughly as possible with a partner so that we could gain insight into how they made their decisions. We collected data in the form of student discussion on Zoom rather than written explanations to maximize discourse around their rationales. Once they finished discussing with their partner, we gathered the whole group and debriefed their responses together, using Zoom to record.

### 2.3.2. Teacher Data Collection

During the writing institute, the first author facilitated a sixty-minute workshop over Zoom. He presented teachers with a Google Doc version of the same ranking activity described above. The instructions for the activity (attached to the same Google Doc as the examples), are shown in Figure 1 below:

**Instructions:** After you have read and discussed…
- **Rank the examples** based on how much you think the student *learned* about writing.
- **Explain** to your partner **why** you ranked the examples as you did. **Take turns**.

As a reminder, here is how each student used ChatGPT:

| Student | What did they ask ChatGPT to do? |
|---|---|
| A | Write the three first sentences |
| B | Make an outline |
| C | Fix their paragraph |
| D | Provide counterarguments |

- Student _______ learned the most about writing.
- Student _______ learned the second most about writing.
- Student _______ learned the third most about writing.
- Student _______ learned the least about writing.

Explain your rankings:

**Figure 1.** Instructions given to both sets of participants.

Next, the first author assigned pairs of teachers to Zoom breakout rooms. As with our student participants, we assigned teachers to discuss and rank the examples in pairs to maximize discourse and gain insight into their rationales. Working on the Google Doc provided, teacher pairs discussed the examples, then ranked them first based on how much they thought each student learned and again by how much the students cheated. Once teachers completed their rankings, the whole group returned to the main Zoom room to discuss their rankings and rationales.

In addition to recording the whole group discussion, we also collected several forms of written data, including teachers' posts in the Zoom chat during the whole group discussion, teachers' annotated comments on the Google Doc, and teachers' post-workshop reflections on a feedback form.

We have summarized our data sources for both groups in Table 1 below.

**Table 1.** Summary of data sources for each group of participants.

| Media | Data Source | Teachers | Students |
|---|---|---|---|
| Text | Rankings (Google Docs) | n = 16 | n = 12 |
| | Chat transcript (Zoom) | n = 1 | n = 0 |
| | Written explanations (Google Docs) | n = 8 | n = 0 |
| | Session feedback forms | n = 16 | n = 0 |
| Video | Paired discussion of rankings and rationales | n = 0 | n = 6 |
| | Whole group discussion of rankings and rationales | n = 1 | n = 1 |

*2.4. Data Analysis*

We used Zoom transcription services to generate a first draft of all video-based data, and we assigned each participant a pseudonym and an alphanumeric code to anonymize the data. Then, we manually reviewed the transcripts, correcting for errors in the automated transcription. As Bucholtz [36] notes, representing discourse in transcription is an inherently ideological act that involves interpretive choices. In our analysis, we used a balance of "naturalized" and "denaturalized" transcription processes [36] to represent participants' speech accurately while also considering the readability of the transcript. We achieved this by retaining as much of participants' original speech as possible while removing distractors like repeated filler words (e.g., "um", "like").

The first author collaboratively coded teachers' written explanations of their rankings (ten Google Docs) with another graduate student. They used a combination of inductive and deductive coding [37], focusing on the implicit and explicit criteria participants used to determine the degree to which each student learned and cheated with ChatGPT. Following Campbell [38], they determined a meaningful unit of analysis, which they agreed would be an explanation for a single ranking. Then, they constructed a preliminary codebook. Using this codebook, the first author coded three additional files (transcripts of two whole group discussions and the teachers' Zoom chat transcript). Then, he and the other graduate student conducted an inter-rater reliability (IRR) test on 58 excerpts using Dedoose [39] qualitative coding software. Their pooled Kappa of 0.68 was not satisfactory, so they resolved discrepancies through a process of negotiated agreement [38] until they reached 100% agreement. They also discussed the codes for which reliability was lowest, resolved discrepancies, and finalized the codebook with more instructive descriptions. The first author coded the remaining transcripts, then conducted a final IRR test with his coding partner. They achieved a satisfactory pooled Kappa of 0.91 and again resolved remaining discrepancies. Finally, the first three authors conducted a thematic analysis [37] of the data.

Once we completed our analysis, we performed a member check [40] with three teacher participants to assess the trustworthiness of the findings. We were unable to conduct a member check with our student participants, who had moved onto different teachers, grades, and schools by the time we completed our analysis.

*2.5. Limitations*

This study's design and methods come with limitations. First, our small sample size (teachers = 16, students = 12) limits the generalizability of findings. However, we used purposive sampling for heterogeneity [33] to recruit participants whose diverse experiences and expertise were directly relevant to our research questions. Additionally, qualitative studies with narrow objectives can reach saturation at relatively small samples [41]. A second limitation is that we used a different suite of data collection methods for teachers and students. We collected student data in person and teacher data on Zoom, which came with constraints

(e.g., we could not record individual breakout rooms). To mitigate this and supplement the Zoom interview data, we collected additional written data from teachers by asking them to provide written explanations for their rationales and recording comments from the Zoom chat. To encourage rich discourse from our student participants, we also privileged verbal explanations over written explanations, which we felt may have inhibited robust explanations.

## 3. Results

### 3.1. Finding 1: Teachers and Students Used Similar Criteria to Describe Learning and Cheating with ChatGPT

Teachers and students used similar criteria to describe learning and cheating with ChatGPT, and they cited them at similar rates to justify their rankings. They considered broad criteria like whether ChatGPT or the student was taking on the cognitive demand, and specific criteria like which specific tasks students offloaded to ChatGPT. To assess relative levels of learning and cheating, they also used similar reasoning strategies like comparing the examples to ways students might get support from non-AI resources. In our analysis, we coded 292 explanations (our unit of analysis) and consolidated these criteria into six themes around how participants described learning and cheating with ChatGPT. We also identified two additional themes that were not criteria but rather strategies they used to rank the examples, which are summarized with descriptions and example quotes in Tables 2 and 3 below:

**Table 2.** Themes for how participants described learning and cheating with ChatGPT.

| Theme | Description | Example Quote |
| --- | --- | --- |
| CognitiveLift | Attending to whether AI or student is doing the "thinking" (e.g. generating ideas, evaluating, analyzing) | *[Student] B [outline prompt] cheated the most because ChatGPT did the most intellectual work for them - how to organize an essay, what arguments to have, etc. Student B did not have to generate ideas for writing. Writing is like a puzzle you piece together and with [generative AI] it does most of the work.* |
| Models | Seeing models or examples and transferring new knowledge to their own writing. | *Student B [outline prompt] learned the most about writing because a lot of students don't know what an outline looks like or how to structure their ideas.* |
| Edit | Attending to corrections (e.g., grammar, punctuation, spelling). | *Student C [edit prompt] learned the second most about writing. They had a starting point, but learned about grammar and word choice in a way that they could probably apply to future writing.* |
| AltPerspectives | Considering perspectives different from the students' own. | *[Student] D [counterargument prompt] cheated the third most because they used ChatGPT to research counterarguments, but didn't use it for the actual argument.* |
| GettingStarted | Getting started independently or moving from nothing to something. | *Student A [sentences prompt] \*possibly\* learned the third most because they are using ChatGPT to get started, and with that start, they may be able to write the rest.* |
| StyleVoice | Attending to writers' style, voice, personal experiences, and creativity. | *We chose [Student] D [counterargument prompt] second [most learned] because it helps students elevate their style.* |

**Table 3.** Strategies participants used to determine whether students were learning or cheating with ChatGPT.

| Theme | Description | Example Quote |
| --- | --- | --- |
| Context | Considering contextual factors such as the assignment goal, student population, or what the student did with ChatGPT's output. | *Depends on the assignment. If the assignment was to write an outline, this is 100% cheating.* |
| Comparison | Comparing student use of ChatGPT to an existing resource, either digital, analogue, or human. | *Student C [edit prompt] cheated the least because they already had their paragraph, but they just asked ChatGPT to edit it—kind of like a spell checker.* |

Teachers and students generated similar criteria, including the use of ChatGPT to provide models, offer alternative perspectives, and edit writing. They also cited these criteria at similar rates. For example, both groups' most frequently cited criterion was "cognitive lift", which we defined as attention to whether the student or ChatGPT was doing more of the "thinking". Teachers cited this criterion in 23% of their explanations, and students cited it at 27% of theirs. Figure 2 shows the frequency of each theme as a percentage divided by the total number of explanations coded, some of which were double-coded into more than one theme.

### Theme frequency in teacher and student explanations

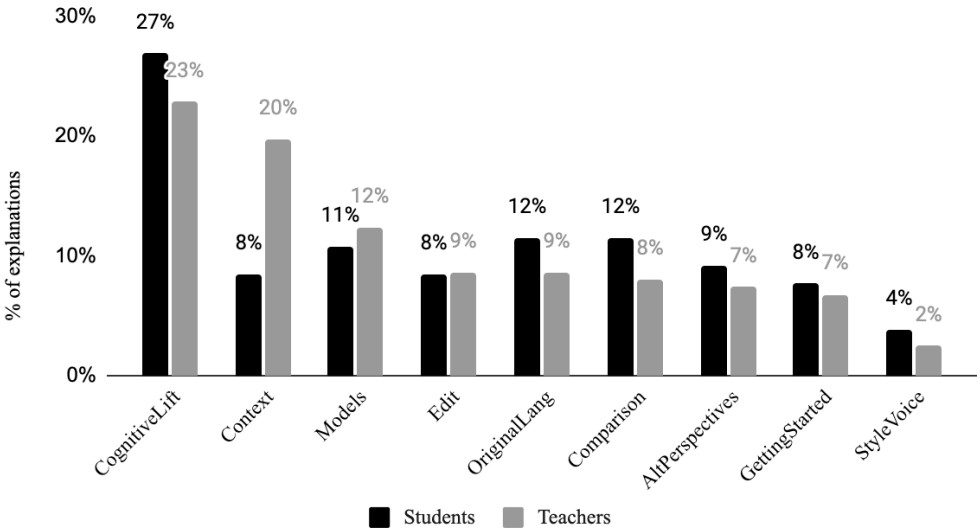

**Figure 2.** Frequency of each theme cited in teacher and student explanations about learning and cheating with ChatGPT. All participants provided explanations for each example.

Importantly, teachers more frequently cited the idea that how much a student cheated or learned depends on contextual factors such as the assignment objectives, individual student goals, or what the student did with ChatGPT's output. In attempting to rank four examples of how a student might use ChatGPT in a writing assignment, teachers cited context at over twice the rate (19.5%) as students (8.6%). Upon deeper analysis, this margin revealed several important tensions in how both groups defined cheating and learning with ChatGPT. We focus on these tensions in Finding 3.

### 3.2. Finding 2: Students and Teachers Had Similar Opinions about Learning with ChatGPT but Different Opinions about Cheating with ChatGPT

Teachers and students assigned similar rankings to which student learned the most. Among both groups, the largest percentage of participants ranked Student A (sentences prompt) as learning the most, followed by Student B (outline prompt), then Student C (edit prompt), and finally Student D (counterargument prompt). Figure 3 shows a similar distribution in how teachers and students ranked which student learned the most, along with a quote from each group justifying their ranking.

By contrast, we found wider variation between groups in ranking which student cheated the most. Figure 4 shows how participants ranked which student cheated the most, accompanied by a quote from each group justifying their ranking.

Nearly half (45%) of the students stated that Student D (counterarguments prompt) cheated the most, while none of the teachers did. Additionally, a closer examination of how teachers ranked Student A (sentences prompt) highlights more within-group variation; 44% of teachers said Student A learned the most, while 56% said they cheated the most. This split illustrates, perhaps counterintuitively, that participants did not view learning and cheating as opposing processes. Participants expressed similar conflicting opinions about nearly all the examples. One teacher even wrote next to her ranking, "Ha! I am

surprised by my own ranking: Learned the most and cheated the most are the same (D, counterarguments prompt)!" We focus on these tensions in the final finding.

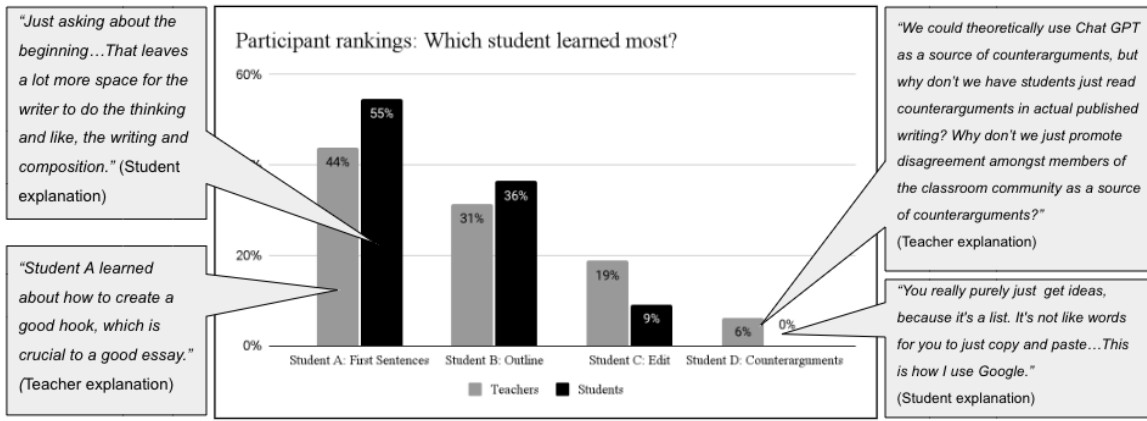

**Figure 3.** Comparison of how teachers and students ranked which student learned the most, accompanied by a representative quote from each group.

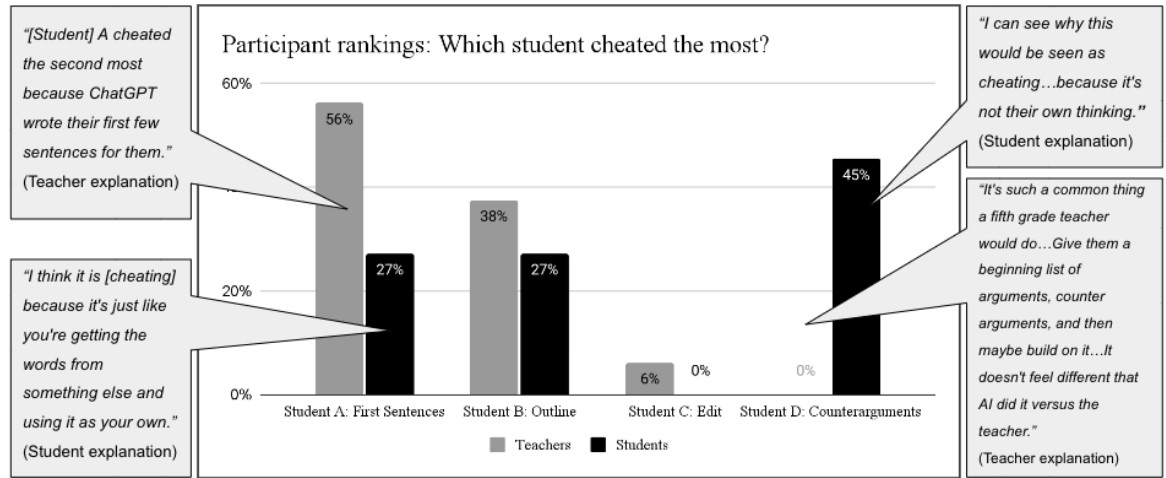

**Figure 4.** Comparison of how teachers and students ranked which student cheated the most, accompanied by a representative quote from each group.

### 3.3. Finding 3: Variation within and between Groups Centered on Four Recurring Tensions

Teachers and students frequently explained that the degree to which the example student learned and cheated depended on context, including the goal of the task and how the student used ChatGPT's output. We captured this theme using the code Context and concluded that variation in how participants ranked the examples was largely due to four tensions:

1.  Using ChatGPT as a shortcut versus as a scaffold;
2.  Using ChatGPT to generate ideas versus language;
3.  Getting support from ChatGPT versus analogous support from other sources;
4.  Learning from ChatGPT versus learning as a whole.

When we saw differences among participants in their assessment of cheating, their differences in opinion were often related to one or more of these tensions. Their differences also provide valuable nuance in how teachers and students might construct shared norms and understandings of responsible school use of tools like ChatGPT, which we will discuss in the final section.

### 3.3.1. Tension 1: Using ChatGPT as a Shortcut versus as a Scaffold

The first tension involved the distinction between using ChatGPT as a shortcut versus as a scaffold. Some teachers assumed—implicitly or explicitly—that the student would use ChatGPT's output with little or no evaluation or citation. In this scenario, the student would shortcut their own learning. Others raised the possibility that the student might use ChatGPT as a scaffold, critically evaluating its output and making deliberate rhetorical choices about which parts to adopt, modify, or omit. In Finding 2, we described how nearly half of the students ranked Student D (counterargument prompt) as cheating the most, while none of the teachers did. This example yielded conflicting rankings within groups as well. Nearly half of the students thought Student D cheated the most, while two thought they cheated the least. Similarly, one teacher thought Student D learned the most, while others thought they learned the least. The differences in opinions on Student D were primarily based on different assumptions about whether the student used ChatGPT's output as a shortcut or as a scaffold. For example, the student quoted in Figure 4 above explained, "These are the seven counterarguments. Now I can see why this would be seen as cheating, because it's not their own thinking". Another countered, "I feel like this question sounds like they're trying to cheat but if you really look at it, I think this is like the least cheating to me. Because I feel like this is a way that, like you really purely just get ideas... This is a list for me to look at, and I think this is a solid way to use it". The first student viewed ChatGPT as doing the thinking for Student D; the second student viewed it as supporting Student D's thinking by providing ideas for them to reference and build on. One teacher summed up both perspectives in this tension, explaining:

The student asked ChatGPT to provide counterarguments, right? And on the one hand... That actually is forcing the student to think of arguments or rebuttals to the counterargument, which is learning, right? But if that student then copies and pastes the counterargument from ChatGPT into their essay, and doesn't acknowledge where the counterargument is coming from, or tries to pass it off as a hypothetical counterargument that their own brain thought of, then that becomes 'cheaty'.

The wide variation in opinions about counterarguments, both between and within groups, underscores the importance of contextual factors that influence both processes, such as assignment objectives and how the student used ChatGPT's suggestions. Both groups' rankings depended largely on whether they believed Student D copied ChatGPT's suggested counterarguments or evaluated them critically. Some saw ChatGPT as doing most of the cognitive work by generating ideas, and they ranked Student D as learning less and cheating more. Others saw ChatGPT's suggestions as a way to introduce students to alternative perspectives and strengthen their argument.

### 3.3.2. Tension 2: Using ChatGPT to Generate Ideas versus Language

Both groups also made a distinction between using ChatGPT to generate ideas and using it to generate specific language. This tension was most clear in the example of Student A, who asked ChatGPT to suggest two possible first sentences for an essay. Teachers and students varied considerably in how they ranked this example, with 56% of teachers ranking Student A as cheating the most, while 45% of students ranked them as cheating the second least. Again, however, there was variation within each group as well. In considering the degree to which Student A cheated, one student said, "I think it is [cheating] because it's just like you're getting the words from something else and using it as your own". Another saw it differently: "I don't feel like it is necessarily cheating because it's getting the person to think about it...it's not necessarily writing the whole thing, it's just helping the person think about it". Where the first student saw this example as a clear case of plagiarism, the second saw it as a scaffold, noting that, "It also doesn't mean that the person is going to use either". Teachers expressed similar conflicting views, with one asserting, "A cheated the second most because ChatGPT wrote their first few sentences for them", and another saying, "The first 3 sentences, if in the context of discussing what is a good hook, is a

different situation than if the student then copies and pastes one of those sentences into their own essay". A different teacher summarized this tension as follows:

> Yeah, I think the question is, like, what do we consider to be more 'cheaty'? Is it taking someone else's idea or taking someone else's words or language? Which I don't really know the answer to. I'm more inclined to have kids come up with their own ideas. But then sometimes you need to give them an idea as a scaffold.

Figure 5 shows how participants ranked the degree to which Student A (sentences prompt) cheated, along with a representative quote from one teacher and one student.

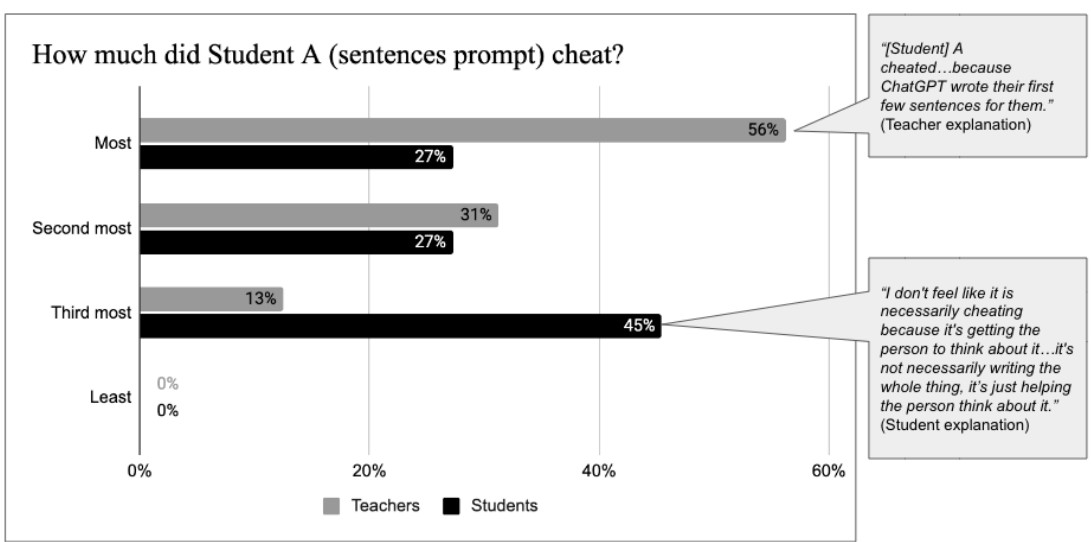

**Figure 5.** Participants' rankings of how much Student A (sentences prompt) cheated and representative explanations.

Participants expressed similar ambivalence with other examples we asked them to rank. For instance, one teacher compared Student A (sentences prompt) to Student B (outline prompt) and concluded, "I feel like it's a different kind of cheating. The other one was, like, write my topic sentence. And then this one's more like, here's a structure for critical ideas, but you have to articulate the ideas yourself. It's a little different to me". It is worth noting that this second tension between using ChatGPT to generate ideas versus language overlaps with the first tension between using ChatGPT as a shortcut or as a scaffold. Teachers and students were generally in agreement that copying and pasting suggestions from ChatGPT without citation constituted cheating.

3.3.3. Tension 3: Getting Support from ChatGPT versus Analogous Support from Other Sources

One way that both groups of participants arrived at their opinions about learning and cheating was by comparing the support the hypothetical student received from ChatGPT with analogous forms of support they might receive from non-ChatGPT sources, including digital resources, peers, or a teacher. In most instances, when they identified a non-ChatGPT resource that could provide the same support, they were less likely to view the behavior as cheating. The most frequent non-ChatGPT resources they cited included Google, editing tools (e.g., Grammarly, spell-check), and teachers themselves. Students applied this comparison tactic to Student D, who asked for counterarguments ("I feel like this is just like asking a teacher like and teachers give you ideas like this. . .I think this is optimal learning".) and Student C, who asked ChatGPT to edit a paragraph ("So. . .kind of like how you have Grammarly, it's kind of like that but using ChatGPT"). Teachers made almost identical comparisons for Student D ("The very common thing a fifth grade teacher would do would [be to] give them a beginning list of kind of arguments, counterarguments, and then maybe build on it. So it feels so common to me. . . It doesn't feel different if AI did

it versus the teacher".) and Student C ("C was mostly just a replacement for Grammarly".). One teacher made a connection between their familiarity with technology and acceptance of students' use of it. She explained, "We agreed that editing assistance was not cheating. This is interesting because we have adapted to editing software already. So becoming familiar with a tech solution is gradually more accepted". Interestingly, there were no instances in which either teachers or students made a comparison that increased their perception of how much the student cheated.

### 3.3.4. Tension 4: Learning from ChatGPT versus Learning as a Whole

One final tension raised by students specifically—but not teachers—was the tension between learning from ChatGPT versus learning as a whole. A small group of students argued that Student C, who only asked ChatGPT to edit their essay, learned the most. When we asked them why, one replied, "Because it was their paragraph". A second student added, "They put their information there, and it was more so like swapping words or taking out words rather than asking [ChatGPT] can you give me counterarguments". This group of students viewed editing as the least cognitively demanding part of writing; thus, they reasoned that most of Student C's learning took place before using ChatGPT to help them edit. Another student explained, "I think mainly it's like, thinking for yourself, building your problem-solving skills on your own...because...it'll make you learn more if you learn it yourself instead of using a tool". This idea did not surface frequently, and it did not surface at all in teachers' rankings or discussions. Yet, the last student's argument—that "you learn more if you learn it yourself instead of using a tool"—is a provocative one. It highlights the often thin line between using technology to scaffold the learning process and misusing it to lower the cognitive demands of an assignment.

We have summarized the four tensions and included a representative quote for each in Figure 6 below.

**Figure 6.** Four tensions in participants' descriptions of learning and cheating with ChatGPT.

### 4. Discussion

In this study, we sought to examine tensions in teachers' and students' perceptions of learning and cheating with ChatGPT. To achieve this, we used a ranking activity to elicit criteria they used to define learning and cheating. We arrived at three main findings. First,

teachers and students used similar criteria to determine their rankings. Second, teachers and students arrived at similar conclusions about learning with ChatGPT but different conclusions about cheating. Third, disagreement between and within groups centered on four main tensions: (1) using ChatGPT as a shortcut versus as a scaffold; (2) using ChatGPT to generate ideas versus language; (3) getting support from ChatGPT versus analogous support from other sources; and (4) learning from ChatGPT versus learning as a whole.

These findings support prior research that highlights the challenges of defining cheating with respect to writing [15], technology use generally [13, 14], and ChatGPT specifically [22]. Participants in the study illustrated that a use case like asking ChatGPT for counterarguments could either support learning or be considered cheating depending on contextual factors such as the assignment objectives and how students used ChatGPT's output. Participants discussed scenarios in which students might improve their learning by evaluating ChatGPT's suggestions or using them to deepen their own thinking. The possibility that students can use ChatGPT this way supports calls to develop AI literacy frameworks that include teaching students to effectively use generative AI tools [24].

This study contributes to the existing body of research by identifying some of the tensions that may result from different interpretations of general frameworks for responsible AI use. For example, participants had a wide range of opinions (both between and within groups) about whether or not asking ChatGPT for counterarguments constituted cheating. Some viewed this as a legitimate way to consider different perspectives, similar to reading an article or having a discussion with a teacher. Others viewed it as a cognitive shortcut, claiming that "it's not their own thinking". These different perspectives complicate how schools might implement frameworks like the ISTE [19] guidelines for responsible use of intellectual property. However, they also provide opportunities for rich discussion between teachers and students to interrogate crucial questions about how to use technology to support rather than undermine learning.

While our findings do not offer a definitive definition of cheating with ChatGPT or a comprehensive list of behaviors that constitute cheating, they do provide useful heuristics that may help teachers and students arrive at shared understandings. For example, one heuristic participants used to determine whether or not a student was cheating was by considering the goal of an assignment and then determining who was doing the cognitive lift ("Depends on the assignment. If the assignment was to write an outline, this is 100% cheating"). To determine who was doing the cognitive lift, participants used another heuristic by speculating how the student could have used ChatGPT's output ("That actually is forcing the student to think of arguments or rebuttals to the counterargument, which is learning, right? But if that student then copies and pastes the counterargument from ChatGPT into their essay... then that becomes 'cheaty'".) Because concepts like cheating are difficult to define, these types of heuristics can serve as guidelines that teachers and students can reference in determining whether a particular use of AI supports or circumvents learning. In fact, teachers may find heuristics even more useful than "ideal reasoning rules" [42] like strict definitions because they can apply them flexibly in different situations.

This study also contributes to existing scholarship by engaging teachers and students in dialogue and the co-construction of ideas together. Our constructivist approach made space for diverse ideas to emerge and philosophical stances about teaching and learning to be clarified or refined. The range of (sometimes conflicting) beliefs that individuals have about cheating underscores the need for clear and transparent policies. Indeed, tension four—between learning with ChatGPT and learning without it—underscores the risk that "Cognitive outsourcing will become the norm" [43] unless teachers and students create shared agreements about which tasks should and should not be outsourced.

Yet, the presence of policies alone is often insufficient, since many students do not read them [17]. Because of this, we recommend that teachers co-construct norms and policies around AI use with students. They could achieve this by presenting students with real or hypothetical examples of how they might use ChatGPT that they could then discuss with students. The results of this study suggest that students and teachers might begin

with different assumptions about learning and cheating with AI, and that the process of interrogating points of tension together might help them arrive at greater agreement around what constitutes responsible use. Thus, the co-construction approach that we recommend has two benefits: strengthening agreement and ensuring that students know what they are agreeing to. Additionally, consistent school and district-wide policies are equally as important as classroom policies, so administrators might consider taking a similar approach to developing policies with teachers. They might also encourage opportunities for students to explore AI tools under the supervision of their teachers, creating conditions for productive learning. These recommendations should be part of a broader strategy that includes consulting respected style guides for citing AI and aligning with existing policies around academic integrity and technology use.

Our teacher participants provided overwhelmingly positive feedback on the ranking activity and discussion, and during the school year following the study, several even enacted the ranking activity with their students. In a feedback survey, one wrote:

> I have been feeling very nervous about the ways that generative AI will change the teaching of writing, or how I will manage it as a teacher. I thought the activity was so thought provoking because it really pushed us to think about what we consider as "cheating", which is really the anxiety that so many of us have. The blurry lines between cheating and learning about writing was a new idea for me, and something I will continue to think about. I want to do this exact activity with my department, and I think the idea of doing it with students is incredibly intriguing.

Finally, our study supports an approach to AI literacy that shifts away from a "policing" mindset focused on mitigating and punishing cheating and instead towards a mindset that centers student learning [27]. In a feedback survey, a teacher participant wrote, "The main takeaway that I'm left with is that sniffing out cheat-y behavior is very different than sniffing out learning. My initial reaction was to think about cheating, but really, my job hasn't changed—it's about the learning". The themes and tensions we identified through participants' explanations, particularly around cognitive lift, support the view of writing as a mode of learning [44] and privilege the writing process over the final product. Additionally, they underscore the need for teachers to align assessments with learning goals and communicate those goals to students.

Co-constructing norms around cheating is especially relevant to equitable AI literacy. Research has long shown that Black, brown, and indigenous youth are often viewed through the lens of deficit thinking [45], which may lead to an overemphasis on preventing and punishing cheating with AI in schools with high minority enrollment. Deficit mindsets and carceral attitudes may also exacerbate discipline disparities that exclude students of color and economically disadvantaged students from educational opportunities [5,6], including opportunities to use AI to enhance learning. By contrast, schools serving more affluent students may be able to fund AI literacy curricula, innovative tools, and staff to support implementation. With these resources, students can learn to use AI tools to accelerate learning, and teachers can use them to increase productivity or redesign assessments to align with the skills students will need to succeed in an evolving high-tech society. Thus, shifting discourse and practice from cheating to learning is a crucial component of equitable AI literacy. Trucano [46] warns that AI may fuel "the next digital divide", which he describes as follows:

> The first digital divide: The rich have technology, while the poor do not. The second digital divide: The rich have technology and the skills to use it effectively, while the poor have technology but lack skills to use it effectively. The third digital divide? The rich have access to both technology and people to help them use it, while the poor have access to technology only.

Drawing a parallel to the debate over book bans, Harrison et al. [47] advocate that educators adopt a "restorative" approach to ChatGPT that places trust in teachers and stu-

dents to engage critically with AI technologies. But unlike book bans, AI bans are virtually impossible to enforce in a digital ecosystem where AI is embedded within applications, like Google Docs, YouTube, and Grammarly, that teachers and students use every day. AI is now ubiquitous in digital experiences, underscoring the need for teachers and students to engage in a dialogue about shared norms and responsibilities.

## 5. Conclusions

As AI technology continues to evolve at an unprecedented pace, so too do norms around what it means to learn and cheat with AI. More research needs to be conducted to better understand how teachers and students develop their perceptions about AI and how schools can operationalize policies into the complex realities of the classroom. To support a shift away from cheating and toward learning, teachers can invite students into dialogue about how to use AI to enhance their learning. They can also redesign assessments to better align with creative and critical thinking skills that are uniquely human. School leaders can support professional learning that provides opportunities for teachers to grapple with tensions within AI literacy. They can create conditions for teachers to engage with new technologies, reflect on student work or other data, consider their own classroom contexts, and be in conversation with others. Administrators can also create consistent policies for responsible use of AI, being intentional about which apply across the institution and which are better left to the discretion of individual teachers. Doing so requires some degree of shared understanding among stakeholders—including teachers and students—about the benefits and risks of AI usage. These recommendations represent a partial but important blueprint for developing effective AI literacy for a rapidly changing world.

**Author Contributions:** Conceptualization, C.M., H.W., S.L., S.W.B. and J.P.; methodology, C.M., S.L., S.W.B. and J.P.; resources, C.M., H.W., S.L., S.W.B. and J.P.; investigation, C.M., H.W., S.L., S.W.B. and J.P.; formal analysis, C.M., H.W. and L.P.; data curation, C.M., H.W. and L.P.; writing—original draft preparation, C.M., H.W. and L.P.; writing—review and editing, C.M., H.W., L.P, S.L., S.W.B. and J.P.; validation, and C.M.; visualization, and C.M.; supervision, S.L. and S.W.B.; project administration, C.M., and H.W.; funding acquisition, and C.M. All authors have read and agreed to the published version of the manuscript.

**Funding:** This research received no external funding.

**Institutional Review Board Statement:** This study was approved by the Institutional Review Board of STANFORD UNIVERSITY (protocol code 69423, approved May 2023).

**Informed Consent Statement:** Informed consent was obtained from all subjects involved in the study. Written informed consent has been obtained from the participants to publish this paper.

**Data Availability Statement:** Please contact the first author for information about data availability.

**Acknowledgments:** The authors express sincere gratitude to the teachers and students who shared their time and insights for this study. Publication made possible in part by support from the Berkeley Research Impact Initiative (BRII) sponsored by the UC Berkeley Library.

**Conflicts of Interest:** The authors declare no conflicts of interest.

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
