# Peer review of "Beyond CheatBots: Examining Tensions in Teachers’ and Students’ Perceptions of Cheating and Learning with ChatGPT"

_education, doi:10.3390/educsci14050500_

Round 1
Reviewer 1 Report
Comments and Suggestions for Authors
This is a very interesting paper indeed. This is a very important and timely subject and it is a potentially significant contribution to the current research on the topic of AI in education. The focus of the manuscript and the objective of the research are clear already from the abstract. The manuscript is well structured and the references are sound, most of the references are very current. In the reference list there seems to be only one paper on AI literacy, which is surprising considering that it is one of the main topics of this paper. Hence, the paper would benefit greatly from more references on AI literacy. Overall this is an excellent paper. The tables and figures are very good and informative!
Author Response
Thank you for your feedback. We have added the following paragraph about AI literacy (with citations) in Lines 40-49 of our introduction:
Attempts to define AI literacy predate ChatGPT. Long and Magerko (2020) define AI literacy as, “a set of competencies that enables individuals to critically evaluate AI technologies; communicate and collaborate effectively with AI; and use AI as a tool online, at home, and in the workplace.” Other researchers have avoided definition and instead proposed broad frameworks for AI literacy. Ng et. al. (2021) reviewed thirty peer-reviewed articles and identified four aspects for fostering AI literacy: understanding AI concepts, applying computational thinking, designing and creating with AI, and attending to AI ethics. Similarly, Heydar and Possega (2021) characterized AI literacy as having three dimensions: functional/technical literacy, critical/cognitive literacy, and sociocultural/socioemotional literacy. Irrespective of conceptualization, defining responsible use of AI is a crucial part of any framework for equitable AI literacy (Lee et al., 2021).
Reviewer 2 Report
Comments and Suggestions for Authors
I consider it an important study and a call to attention to the current problem, the importance of using ChatGPT and the perspective of students and teachers on its use in learning: a learning support tool or a promoter of cheating?
I consider only two aspects desirable to improve the paper, namely:
- the introduction of a reference to the Dedoose tool
- the addition of some general considerations about how schools might implement frameworks like the ISTE (2016) guidelines for responsible use of intellectual property.
-Is it possible to describe the general background of each student (freshmen, sophomores, juniors and the senior one)?
Author Response
Thank you for your feedback. We have made the following revisions based on your feedback.
- the introduction of a reference to the Dedoose tool
We have referenced Dedoose in-text in Line 269 and in our References section in Line 730.
- the addition of some general considerations about how schools might implement frameworks like the ISTE (2016) guidelines for responsible use of intellectual property.
This piece of feedback was useful, but we felt it was slightly out of scope of this paper, which focused primarily on students and teachers rather than school or district implementation. The feedback was instructive, though, and we added the following section in Lines 568-573 in order to connect our study to attempts at broader policy implementation:
Additionally, consistent school and district-wide policies are equally as important as classroom policies, so administrators might consider taking a similar approach to developing policies with teachers. These recommendations should be part of a broader strategy that includes consulting respected style guides for citing AI and aligning with existing policies around academic integrity and technology use.
-Is it possible to describe the general background of each student (freshmen, sophomores, juniors and the senior one)?
We did not collect data about individual students beyond what we reported, so added the following sentences in Lines 158-160 to provide more context:
These students represented the broader school demographics and varied in their English class grades. Some were familiar with ChatGPT and others had never heard of it.
Reviewer 3 Report
Comments and Suggestions for Authors
The article addresses a very current and interesting topic, so it should be considered for its publication. Furthermore, it presents a well-founded theoretical framework supported by very recent bibliography. However, it must be improved in the following aspects:
1) It is necessary to better define the methodology developed. The authors describe the process, but do not clearly refer to the type of methodology used, supporting it with adequate references.
2) Regarding the methodology, it is pertinent in relation to the topic, but the qualitative methodology requires arguing and describing the entire process with great rigor, therefore, it is necessary for the author/s to explain whether the recording of the interviews was carried out. in a reliable manner and respecting the criterion of stylistic literality, following Pujadas (2000). They should also mention whether a process of validation of the information by the informants through feedback from the participants has been followed. Additionally, they must explain how each participant has been coded to protect the confidentiality of the professionals interviewed.3) It is necessary to review the references, adapting them all to the Regulations of the journal.
Author Response
Thank you for your helpful feedback. Here are the revisions we made in response:
Methodology:
- In Lines 149-151, we added the following statement to more explicitly ground our description of our methods:
- Following Saldaña (2011), we used fundamental qualitative methods, relying on group discussion grounded in a textual artifact to elicit participants’ ideas around the topic of learning and cheating with AI.
Transcription practices:
- Thank you for the suggested reference. We followed Bucholtz (2000) in our transcription practices and added the following paragraph to more thoroughly describe our practice and to address your comment around participant confidentiality:
- We used Zoom transcription services to generate a first draft of all video-based data, and we assigned each participant a pseudonym and an alphanumeric code to anonymize the data. Then, we manually reviewed the transcripts, correcting for errors in the automated transcription. As Bucholtz (2000) notes, representing discourse in transcription is an inherently ideological act that involves interpretive choices. In our transcription process, we used a balance of “naturalized” and “denaturalized” transcription processes (Bucholtz, 2000, p. 1461) to represent participants’ speech accurately while also considering the readability of the transcript. We did this by retaining as much of participants’ original speech as possible while removing distractors like repeated filler words (e.g. “um,” “like”).
Member Checking:
- We added the following paragraph in Lines 278-281 to describe our data validation with participants:
- Once we completed our analysis, we performed a member check (Doyle, 2007) with three teacher participants to assess the trustworthiness of the findings. We were unable to conduct a member check with our student participants, who had moved onto different teachers, grades, and schools by the time we completed our analysis.
References:
- We noted that the journal's formatting guidelines call for semicolons between authors. We have updated our references accordingly.
Reviewer 4 Report
Comments and Suggestions for Authors
The article presents and discusses the results of a qualitative research on teachers’ and students’ perceptions of cheating and learning. Researchers selected Pre-K through post-7 secondary (n=16) and a linguistically diverse group of students (n=12) in a small city of the West Coast. Researchers created examples to represent different uses of ChatGPT and asked participants to rank them in order of how much they thought each student learned and cheated, then to explain their rankings. Teachers and students used similar criteria to determine their rankings. Moreover, teachers and students arrived at similar conclusions about learning with ChatGPT. On the contrary, the two groups reached divergent conclusions about cheating. By analyzing the results of the research, the authors identified disagreements centered around four main tensions: 1) using ChatGPT as a shortcut versus as a scaffold; 2) using ChatGPT to generate ideas versus language; 3) getting support from ChatGPT versus analogous support from other sources; and 4) learning from ChatGPT versus learning as a whole.
The article is well-written. The research is well-designed and provides original and interesting results. However, I would like to suggest three points for reflection and revision to the Author(s).
The first concerns the design of the empirical research. Since it is a case study, it might be useful to provide some additional elements to contextualize the data and results. For example, information about the academic performance of the participating students as well as on the age of the teachers could be relevant.
The second observation concerns the conclusions. The Author(s) propose a "constructivist approach" through which students and teachers can engage in dialogue and together define norms and policies regarding the use of AI. In this part of the article, the Author(s) could delve more into detail about how schools could adopt such an approach and, above all, how the results of this study could concretely contribute to this process.
Thirdly, from the research emerges an interesting result about the tension between learning from ChatGPT versus learning as a whole. Indeed, this tension highlights the advantages and disadvantages of the so-called "cognitive outsourcing" (see, for example Yu, S., Lu, Y. (2021). Prospects and Reflections: Looking into the Future. In: An Introduction to Artificial Intelligence in Education. Bridging Human and Machine: Future Education with Intelligence. Springer, Singapore. https://doi.org/10.1007/978-981-16-2770-5_9) It appears as an interesting fact, moreover, that the students, not the teachers, have brought out this aspect in a somewhat problematic way. In the writer's opinion, it could be useful to explicitly state and argue this point, as it may support the thesis that it is important to co-construct norms and policies in this regard through coordinated and shared processes involving both teachers and students.
Author Response
Thank you for your feedback. Here are the revisions we made in response:
Contextualizing Data:
- We have added the following sentences to provide more context on our participants:
- Lines 158-160 (Students): These students represented the broader school demographics and varied in their English class grades. Some were familiar with ChatGPT and others had never heard of it.
- Lines 178-179 (Teachers): The group included ten middle and high school teachers, three elementary school teachers, three post-secondary teachers, and one pre-K teacher.
School recommendations:
- We appreciate this piece of feedback but felt that school recommendations went slightly beyond the scope of our study, which focused on student and teacher perceptions as opposed to school level implementation. Still, your feedback was instructive, and we revised our discussion section to situate our study and recommendations within the broader context of school and districts' efforts around AI literacy.
- Our main revision comes in Lines 560-573: Because of this, we recommend that teachers co-construct norms and policies around AI use with students. They could do so by presenting students with real or hypothetical examples of how they might use ChatGPT that they could then discuss with students. The results of this study suggest that students and teachers might begin with different assumptions about learning and cheating with AI, and that the process of interrogating points of tension together might help them arrive at greater agreement around what constitutes responsible use. Thus, the co-construction approach that we recommend has two benefits: strengthening agreement and ensuring that students know what they are agreeing to. Additionally, consistent school and district-wide policies are equally as important as classroom policies, so administrators might consider taking a similar approach to developing policies with teachers. These recommendations should be part of a broader strategy that includes consulting respected style guides for citing AI and aligning with existing policies around academic integrity and technology use.
Cognitive Outsourcing:
- We have added the following section to explicitly connect our findings to cognitive outsourcing:
- Lines 554-558:
Indeed, tension four - between learning with ChatGPT and learning without it - underscores the risk that “Cognitive outsourcing will become the norm” (Yu, 2021, pp. 197) unless teachers and students create shared agreements about which tasks should and should not be outsourced.
Reviewer 5 Report
Comments and Suggestions for Authors
The manuscript "Beyond CheatBots: Examining Tensions in Teachers’ and Students’ Perceptions of Cheating and Learning with ChatGPT" addresses how teachers and students define and perceive learning and cheating in the context of using ChatGPT in educational settings. Overall this is a novel and interesting topic, of interest to the readers of Education Sciences. The article includes all expected sections from a research paper reporting empirical findings. The English level is adequate.
However, I have several issues that I recommend the authors to address to improve their research report.
My first concern is related to the research goal, that appears somewhat inconsistently throughout the paper: in the abstract it is mentioned the “exploration of tensions between students and teachers’ views”, on the research goals section it is claimed that the study aims to “explore both the risks and potential of generative AI for young writers”. On the discussion one reads again “we sought to examine what criteria teachers and students used to 460 define learning and cheating with ChatGPT”. The research goal should be consistently presented throughout the manuscript.
Theoretically, this research is grounded in a discourse around responsible use of AI in education. It discusses the evolution of definitions and policies regarding cheating and responsible technology use, particularly focusing on the integration of AI tools in learning processes. The study builds on previous research indicating varied perceptions and definitions of cheating across different educational stakeholders. However, this discussion could be expanded to provide deeper insights into how students and teachers develop their perceptions about AI tools and about learning with AI tools, such as ChatGPT (quite unclear, in my opinion).
The researchers employed qualitative methods, gathering data from two distinct groups: teachers from a professional development institute and linguistically diverse high school students. The study involved activities where participants ranked and discussed different hypothetical scenarios of students using ChatGPT, aiming to parse out the criteria used to define learning and cheating. Data was analyzed using a combination of inductive and deductive coding. According to the description, the authors used a rigorous process to ensure the validity of the coding. However, no measures for intercoder reliability were provided.
Regarding the results, the main findings were reported in a dense and detail way: teachers and students agreed on criteria for using ChatGPT to learn but differed on what constituted cheating, influenced by four tensions: using ChatGPT as a shortcut or scaffold, generating ideas versus language, comparing ChatGPT's support to other sources, and the role of AI in specific learning processes versus overall education.
The paper seems to introduce new quotes in the discussion section that were not explicitly analyzed or referenced in the results section, which could raise concerns about the consistency of data reporting and interpretation throughout the study. The paper concludes that defining cheating in the context of AI use is complex and context-dependent, influenced by factors such as the objectives of the assignment and the ways students interact with AI outputs.
More explicit suggestions for future research could guide subsequent studies, such as exploring specific interventions that might shift perceptions of AI tools from cheating aids to educational enhancers. Additionally, the authors could further elaborate on practical steps that educators and policymakers can take based on these findings. This might include specific strategies for integrating AI tools responsibly into curriculums.
Author Response
Thank you for your feedback. We have made the following revisions to address them:
Inconsistent goals:
- In Lines 502-504, we have clarified our primary goal with our intermediate goal: In this study, we sought to examine tensions in teachers and students perceptions of learning and cheating with ChatGPT. To do so, we used a ranking activity to elicit criteria they used to define learning and cheating.
- In Lines 140-141, we rewrote the sentence you quoted: We wanted to explore teachers’ and students’ perceptions of learning and cheating with ChatGPT.
Insights into how students and teachers developed their perceptions of AI:
- Thank you for this feedback. Respectfully, we felt that this important question was not in the scope of this study. The first author did collect additional data on teachers' perceptions that may be analyzed for a different study. We did not collect this data for our student participants, and therefore could not draw conclusions in this study.
Coder reliability:
- We described our coding process in lines 260-277 and added descriptions around transcription practices and member checking based on another reviewer's feedback.
New quotes in discussion section:
- The quotes introduced in our discussion section came from teacher feedback surveys following the summer institute. These feedback surveys were not part of our analysis, because they did not directly relate to our research questions. However, we thought the quotes provided great insight into what teachers felt was valuable about the exercise, which informed our recommendations.
Practical recommendations:
- We have incorporated your suggestion (and other reviewer's suggestions) to elaborate on our recommendations for educators.
Future research:
- We have added the following sentence into our conclusion in Lines 626-629:
- More research needs to be done to better understand how teachers and students develop their perceptions about AI and how schools can operationalize policies into the complex realities of the classroom.